# School Connectedness and Academic Burnout in Middle School Students: A Multiple Serial Mediation Model

**DOI:** 10.3390/bs14111077

**Published:** 2024-11-11

**Authors:** Hui Zhao, Mengjiao Han, Zhenzhen Wang, Bangdan Liu

**Affiliations:** Faculty of Education, Henan Normal University, Xinxiang 453007, China; zhaohui@htu.edu.cn (H.Z.); 2210283138@stu.htu.edu.cn (Z.W.); 2310283140@stu.htu.edu.cn (B.L.)

**Keywords:** academic burnout, school connectedness, core self-evaluation, autonomous motivation, academic self-handicapping

## Abstract

Higher levels of school connectedness are associated with better study habits, but their relationship with academic burnout and the underlying mechanisms have not been revealed. We used a questionnaire to investigate the relationship between school connectedness and academic burnout and the mediating mechanisms of burnout in a sample of 394 Chinese middle school students, controlling for class, gender, and grade level. The results revealed that (1) school connectedness, autonomous motivation to learn, and core self-evaluations were significantly negatively related to academic burnout; and that (2) academic self-handicapping, core self-evaluations, and autonomous motivation to learn individually mediated the effects of school connectedness on academic burnout and mediated the effects of multiple factors. Therefore, educators should pay attention to the emotional needs of junior high school students, increase the level of school connectedness, consciously help students cultivate positive psychological factors such as autonomous motivation and core self-evaluations, reduce academic self-handicapping, increase their learning pleasure, and alleviate junior high school students’ academic burnout.

## 1. Introduction

As today’s Chinese society demands increasing education, it also imposes heavier academic burdens on junior high school students; these pressures are more likely to lead junior high school pupils to suffer psychological issues such as burnout and weariness [1]. Academic burnout (AB) is caused by large study pressures in the process of learning or a lack of interest in learning, resulting in a chronic, unpleasant state of mind about learning, which manifests itself as weakened motivation to learn, a sense of detachment from learning, and a lack of joy and fulfillment from learning [2]. Research has demonstrated that AB is prevalent among middle school students and increases with grade level [3]. ABs have attracted much attention from scholars not only in China but also worldwide [4]. ABs are considered one of the primary factors influencing the healthy development of teenagers, seriously jeopardizing their physical, mental, and academic development [5,6]. Therefore, the AB problem has also gradually become a topic of concern for educators, and it is necessary to understand the mechanisms of AB in middle school students to decrease AB. Currently, the main factors found to influence AB are family, social support [7], and one’s own domain [8]. However, in the school domain, few studies have emphasized the importance of the school environment and the support of schoolteachers [9] as important influences on AB. However, the majority of students’ activities take place in schools, and factors such as the school environment are easier to control and change than other factors are; additionally, the school environment can be more readily adapted to combat students’ AB by adjusting various factors [10]. Therefore, further analysis of the factors influencing AB from the school perspective not only helps provide an understanding of the reason for AB but also enriches the related theories of AB and provides effective insights for reducing AB.

School connectedness (SC) is the emotional connection that individuals establish with their schools and those in the school environment and is studied by limiting the subject and object to the school context to determine students’ sense of belonging to the school, identifying with the school, and feeling cared for, acknowledged, and supported in the school [11]. Conservation of resources (COR) theory holds that individuals tend to adopt good coping strategies when their resources are sufficient, which provides sufficient guarantees for individuals to cope with ABs [12]. If individual resources are limited and difficult to cope with, persistent resource threats and depletion lead to AB [13]. SC can also provide psychological and social support, and the acquisition and preservation of many resources can help individuals alleviate AB. However, SC in China is unique in that teacher–student relationships, teaching styles, and classroom interactions differ between China and the West [14]. Influenced by traditional Chinese culture, Chinese teachers are respected and loved and have a high status [15], and there is a saying that “once a teacher, always a father”, which shows that Chinese teacher–student relationships have a special connection compared with those in the West. Therefore, it is indispensable to study the relationship between SCs and ABs in the context of China’s educational culture. However, few studies have explored the relationship between the two, and the intrinsic mechanism underlying this relationship has not yet been explored in depth, which is not conducive to revealing the causes of AB among middle school students and restricts the dissemination of the research results to educational practice.

According to COR theory, SC is an external environmental resource, whereas core self-evaluations (CSE), autonomous motivation (AM), and academic self-handicapping (ASH) are internal psychological resources that may have an impact on AB. SC provides a supportive environment that facilitates the replenishment of individuals’ personal capacities, which helps them maintain resources [12]. CSE enables individuals to believe in their own abilities and positively face learning and life, thus, enriching the individual’s store of psychological resources needed for the learning process [16]. AM maintains students’ learning state to achieve self-resource retention [17]. All three of these factors can effectively replenish the resources lost by individuals. In contrast, ASH, as a negative coping style, inhibits the achievement of challenging goals and personal growth enhancement [18], preventing individuals from building general resources. Moreover, based on the psychological mediation framework (PMF), SC, as an external factor, affects the level of AB through internal mediating factors [19]. The internal mediating factors also include risk factors and protective factors; CSE and AM are protective factors, whereas ASH is a risk factor, and they may play an intermediary role between the two. However, previous studies have examined only individual risk or protective factors [20], making it difficult to fully understand the influence path of SC on AB. The cumulative effect model of risk-protective factors suggests that protective factors can play a positive role by weakening risk factors [21]. Therefore, in this study, ASH, AM, and CSE, as risk factors and protective factors, respectively, are included in the internal mechanism of SC affecting AB to explore the differences in the impact of the two and their interaction on AB.

In summary, this study takes Chinese middle school students as the subjects, constructs a multisequence mediating model according to COR and PMF theories, and explores in depth the impact of SC on AB, as well as the multiple sequential mediating roles of CSE, AM and ASH in the above relationships.

### 1.1. Theoretical Background and Hypothesis Development

#### 1.1.1. Theoretical Framework

The use of COR, an important theory for explaining the mechanism of burnout, has been widely accepted by scholars. Among the psychological resources that SCs may bring to individuals, AM and CSE are particularly important [22]. CSE enables individuals to believe in their own abilities and actively face learning and life, enriching the psychological resources needed in the learning process [16]. AM can maintain students’ learning status and realize self-resource maintenance [17]. They can effectively supplement the resources lost by individuals, provide internal psychological resources for coping with academic difficulties, and reduce AB. ASH, as a negative coping style, inhibits the realization of challenging goals and the tendency toward personal growth and improvement [18] and hinders individuals from establishing general resources. When students’ resource consumption is not maintained, emotional exhaustion may occur, and AB may be promoted. On the one hand, SC can develop two protective traits, CSE and AM, by activating individuals’ incentive systems [23,24]. On the other hand, the risk factors for ASHs can also be reduced by improving the affirmation of an individual’s own ability [25]. PMF also explains the mediating effects of CSE, AM, and ASH on CS and AB. According to the cumulative effect model of risk-protective factors, CSE and AM can trigger self-affirmation and self-motivation by activating individual protection systems [26,27] to reduce the level of ASH, thereby reducing the risk of AB.

#### 1.1.2. School Connectedness and Academic Burnout

AB refers to students’ negative emotions and behaviors caused by the consumption of emotional resources in the learning process. Among the factors that influence AB, SC has been shown to affect it [28]. SC can be used as a conditional resource to support students so that resources can be maintained and have an impact on AB [12]. School belonging also has a profound effect on students’ academic development; students who are accepted by their peers and teachers are able to have more fun in their studies and reduce their levels of AB [29]. Teacher support and peer support (a sense of support from school members) have important influences on SC; as individual relationship resources, they may reduce the emergence of adolescents’ AB behaviors by promoting more positive school engagement and increased autonomy in learning [30]. Both domestic and international studies have shown that teacher support is beneficial for reducing or preventing students’ AB [31,32]. Higher perceived teacher support promotes students’ academic engagement and reduces problems related to ABs [9]. Moreover, middle school students’ peer support also has a predictive effect on AB behaviors [31], with greater peer support increasing students’ experiences of joy learning at school and lowering the level of AB. It is inferred that there is a close connection between SC and AB. Thus, this study proposes the following hypotheses.

**H1.** 
*SC negatively predicts AB.*


#### 1.1.3. Mediating Role of Core Self-Evaluations

CSEs are individuals’ generalized and complete assessments of themselves and are the most dominant perceptions of material things, status, values, etc. [33]. In the school environment, the support students experience from teachers and peers helps increase their self-confidence in facing learning activities and solving difficulties, which helps them adjust their self-perceptions and improve their self-evaluations and thus, promotes the formation of CSE [22]. School belonging promotes students’ self-concept and self-worth [23], which improves their CSE. Therefore, students’ SC may be positively related to their CSE. CSE can significantly and negatively predict individuals’ problematic behaviors [34]. From a theoretical perspective, CSE, as a type of positive cognition, makes people believe in their own ability and actively face learning and life [16], enriching the psychological resources needed in the learning process and thus, improving AB. In school, individuals with high CSE tend to show greater academic achievement, have greater confidence and motivation in their studies, and have a lower risk of AB problems [35]. In summary, CSE is protective against AB, with higher CSE being associated with lower levels of AB [36]. It is inferred that SC can help adolescents adjust their self-perceptions, form CSE, and be interested in positively confronting learning activities, thus, reducing the tendency toward AB. Students with positive CSE are more likely to use their psychological resources and act proactively in dealing with challenging situations, which can increase their burnout [37]. Therefore, the present study hypothesizes the following:

**H2.** *CSE mediates the relationship between SC and AB*.

#### 1.1.4. Mediating Role of Autonomous Motivation

AM is a motivation with a high degree of self-integration. AM for learning is the internal recognition and pursuit of meaning in learning activities by individuals motivated by extrinsic factors, and it is the dominant motivation for learning behavior [38]. According to PMF, AM, as an internal individual protective factor, may play a mediating role between the external factors SC and AB [19]. Peer support and teacher support are positively correlated with academic autonomy motivation [24,39], and when individuals have established good relationships with others and are able to feel autonomous in their school environments, their levels of academic autonomy motivation increase [40]. Therefore, SC may positively impact the development of academic autonomy motivation in individual students. In addition, higher AM results in better learning engagement and lower levels of AB [9]. AM for learning empowers students to have more autonomy, which can supplement internal positive psychological resources so that students experience more fun in learning to reduce the level of AB [17]. It has also been suggested that students who perceive that their teachers provide more autonomy support experience more autonomous learning motivation, which, in turn, leads to more frequent flow experiences in learning and, subsequently, to less burnout [41]. Therefore, this study proposes the following hypothesis.

**H3.** 
*AM mediates the relationship between SC and AB.*


#### 1.1.5. Mediating Role of Academic Self-Handicapping

ASH is the behavior of individuals to externalize the causes of failure in learning environments to avoid or reduce the negative consequences of underperformance [42]. ASH, as an internal risk factor, may also play a mediating role between SC and AB. Teachers’ teaching styles and peer relationships are all predictors of ASH [43]. If teachers’ value rankings or peer competition are intense, they can cause students to be overly concerned with each other’s scores and rankings, creating ASHs due to the fear of being perceived as less capable by their peers and teachers, which in turn protects their self-worth by preventing failure [25]. Students with high ASH tend to adopt avoidant learning strategies and have higher levels of AB [44]. On the one hand, self-inhibiting learning strategies can lead to lower academic performance and lower levels of intrinsic motivation over time, ultimately leading to a vicious cycle and higher levels of AB [45]. On the other hand, high self-impediment cases are more complex in their psychological activities, which lead to a constant depletion of energy and emotion in individual students, leaving them with less energy to devote to learning activities and, ultimately, leading to AB [18]. Therefore, SC may reduce the use of avoidance strategies and ASH behaviors, which in turn reduces the level of AB. The following hypothesis is proposed in this study.

**H4.** 
*ASH mediates SC and AB.*


#### 1.1.6. Chain Mediation of Core Self-Evaluation, Autonomous Motivation, and Academic Self-Handicapping

On the basis of the previous literature review, CSE, AM, and ASH mediate the relationship between SC and AB. Low CSE is an important factor that leads to the formation of internal and external behavioral problems [46], and ASH is a problematic behavior generated by individual students themselves and may be closely related to their CSE. Individuals’ cognition affects the occurrence of avoidance behaviors [47]; higher CSE is associated with clearer cognition of students’ own abilities and values, making them less afraid of lowering their self-worth by trying but failing and less likely to adopt the learning strategy of avoiding failure [48]. Therefore, we believe that higher student CSE is associated with lower levels of ASH. AM can reduce students’ self-hindering tendencies by alleviating their concerns about academic failure [27]. Students with high AM for learning focus more on the process of learning than the risks associated with failure when faced with difficult academic tasks and tend to employ positive learning strategies rather than avoiding failure [25]. AM helps students maintain their learning behaviors, helps them overcome various difficulties in the learning process, encourages them to solve problems in a positive and active way and makes them less likely to exhibit self-hindering behaviors [49]. Thus, there is a correlation between AM and ASH. In summary, this study speculated that CSE and AM might weaken AB caused by ASHs and play a chain mediating role between SC and AB. The following hypothesis was proposed in this study.

**H5.** 
*CSE and ASH act as chain mediators in the relationship between SC and AB.*


**H6.** 
*AM and ASH act as chain mediators in the relationship between SC and AB.*


### 1.2. The Present Study

On the basis of COR, this study aimed to explore the relationship between SC and AB. More specifically, SC may not only affect AB through the three mediating pathways of CSE, AM and ASH but also affect ASH through the two pathways of CSE and AM. Finally, it has an impact on AM. CSE, AM, and ASH were included in the relationship between SC and AB, and a multiple sequential mediation structural equation model was constructed. Previous studies have shown that sex and grade are related to the level of AB; the level of AB in males is generally greater than that in females, and the level of AB increases with increasing grade [7]. To reduce the impact of both variables on AB and make the results more authentic, both variables are controlled as control variables. The model diagram is shown in Figure 1.

## 2. Materials and Methods

In this study, a junior high school in Henan Province, China, was selected, and a random questionnaire survey was conducted among the students via a convenient sampling method (nested methods were not used in this study). The questionnaire was administered by a psychologist, the instructions were read by the test subjects, and the questionnaires were collected after the subjects finished answering the questions. A total of 450 questionnaires were distributed, and to control for common methodological bias [50], two rounds were conducted 2 weeks apart: the first querying SC and motivation for AM, CSE, and demographic variables and the second querying ASH and AB. A total of 56 questionnaires that failed to be matched were removed from the pre- and post-tests. After eliminating invalid questionnaires such as incomplete answers and irregular answers (incomplete answers meant that most of the questions were not answered, such as the questionnaire missing at least one variable, while the questionnaire missing only a few questions was retained, and the missing value was processed by the maximum likelihood method), 394 valid questionnaires were obtained, and the questionnaire efficiency was 87.5%. Among them, 190 (48.2%) were boys, and 204 (51.7%) were girls; 184 (46.7%) were in the seventh grade, 97 (24.6%) were in the eighth grade, and 113 (28.6%) were in the ninth grade. The subjects ranged in age from 11 to 16 years (M = 13.6 years, SD = 0.94). This study was conducted in accordance with the Declaration of Helsinki and approved by the Ethics Committee of the Faculty of Education, Henan Normal University. Informed consent was obtained from the students and parents of students under 18 years of age.

### 2.1. Measures

#### 2.1.1. Psychological Sense of School Membership Scale (PAAM)

SC was measured via PAAM, which consists of 18 items categorized into two dimensions: feelings of rejection (e.g., “This school is less likely to accept students like me”) and school belonging (e.g., “I feel as if I were part of the school”). The scale was scored on a 1–6 point scale ranging from 1 “strongly disagree” to 6 “strongly agree”. The scale has been shown to have good reliability in previous studies in China [51]. Because only Cronbach’s alpha is limited for testing the reliability and validity of the scale, internal consistency was calculated via Cronbach’s alpha and McDonald’s ω for the full scale and each factor. Overall, the results indicated that all the Cronbach’s alpha and McDonald’s ω values were greater than 0.70, indicating acceptable reliability. These findings show that the reliability of the scale is acceptable and that the scale is suitable for measuring this subject (see Table 1; the same applies below).

#### 2.1.2. Academic Burnout Scale (ABS)

AB was measured via the ABS, which consists of 16 items divided into three dimensions: emotional exhaustion (e.g., “I often feel exhausted at school”), academic detachment (e.g., “I do not feel like I understand anything anyway, so it does not matter whether I learn or not”), and low achievement (e.g., “I do not experience a sense of accomplishment in my studies”). The scale is scored on a 1–5 point scale ranging from 1 “strongly disagree” to 5 “strongly agree”. The scale has good reliability and validity for use in China [7].

#### 2.1.3. Core Self-Evaluation Scale (CSES)

This scale consists of 10 items, including 4 positively scored questions (e.g., “I believe I can be successful in life”) and 6 negatively scored questions (e.g., “I doubt my ability”). The scale is scored on a 1–5 point scale ranging from 1 “strongly disagree” to 5 “strongly agree”. The scale has been shown to have good reliability in previous studies in China [52].

#### 2.1.4. Academic Self-Regulation Questionnaire (ASQR)

AM uses ASQR measurements developed by Black and Deci and improved by Chen [53,54]. This questionnaire is a revision of the original English Learning self-regulation questionnaire by scholar Chen Xuelian on the basis of the literature. Finally, the self-regulation questionnaire was divided into a self-regulation dimension and an external control dimension with 8 questions each. Most previous Chinese studies use the autonomous regulation dimension of this scale to measure autonomous motivation (for example, “I think actively participating in class is a good way to improve my learning”), which shows good reliability and validity in research on measuring autonomous motivation [39] and is more suitable for measuring the autonomous motivation of Chinese junior high school students.

#### 2.1.5. Academic Self-Handicapping Scale (ASHS)

The ASHS is used to measure ASH among secondary school students. The scale consists of 10 items divided into two dimensions: academic self-handicapping (e.g., “I usually put off doing my homework until the end so that people will think I am smart if I do well”) and claimed academic self-handicapping (e.g., “I say that I am not in a good mood before an exam, but in fact, my mood is not as bad as I say”). The scale is scored on a 1–5 point scale ranging from 1 “strongly disagree” to 5 “strongly agree”. The scale has been shown to have good reliability in China [44].

### 2.2. Data Analyses

In this study, the questionnaire data were statistically analyzed via SPSS 21.0 software. First, the five main variables and their dimensions were analyzed via descriptive statistics; then, correlation analysis between two variables was carried out with the Pearson correlation coefficient. The results of the correlation analysis are shown in Table 2. On the basis of the correlation analysis, model 80 of PROCESS, an SPSS plug-in provided by Hayes [55] (2012), was used to construct multiple serial mediation models, and regression analysis was performed for Models 1 to 4. In Models 1 and 2, we regressed the control variable SC on CSE and AM, respectively; in Model 3, we regressed the control variables SC, CSE, and AM on ASH; and in Model 4, we regressed the control variables SC, CSE, AM, and ASH on AM. The results of the whole regression equation were significant; the results of the regression analysis are shown in Table 3. Finally, the outcomes of the bias-corrected nonparametric percentile bootstrap approach and 95% confidence intervals for the test of multiple serial mediation effects showed that the effect intervals of the various indirect mediation paths did not contain 0, indicating that the mediation paths are all established. These results are shown in Table 4.

## 3. Results

### 3.1. Regression Results

The results of the descriptive statistics and detailed correlation analysis are shown in Table 2. SC was positively correlated with CSE (r = 0.58, *p* < 0.01) and AM (r = 0.58, *p* < 0.01). There was a significant positive correlation between CSE and AM (r = 0.47, *p* < 0.01). AB and SC were positively correlated (r = −0.63, *p* < 0.01). There was a significant negative correlation between CSE (r = −0.78, *p* < 0.01) and AM (r = −0.55, *p* < 0.01). There was a significant negative correlation between ASH and SC (r = −0.37, *p* < 0.01). There was a significant negative correlation between CSE (r = −0.34, *p* < 0.01) and AM (r = −0.38, *p* < 0.01). ASH was positively correlated with AB (r = 0.47, *p* < 0.01). The results did not conflict with the research hypotheses and confirmed that the hypotheses had some reliability.

### 3.2. Structural Model

#### 3.2.1. SC and AB

When class, gender, and grade were controlled, SC was significantly correlated with AB (see Table 3, Figure 2). Specifically, SC was significantly and negatively correlated with AB (β = −0.17, *p* < 0.001), with a 95% confidence interval of [−0.20, −0.07] and an adjusted R^2^ = 0.69, F = 79.61, *p* < 0.001 for the model.

#### 3.2.2. Mediating Role of CSE, AM, and ASH

After controlling for the main effects of class, gender, and grade on students, the mediating variables—CSE, AM, and ASH—were added to the model to obtain the path model shown in Figure 2. All path coefficients reached statistical significance (*p* < 0.01), and none of the path confidence intervals contained zero (see Table 3, Figure 2), nor did the indirect path confidence intervals contain zero (see Table 4). Specifically, CSE mediated the relationship between SC and AB, where SC positively predicted CSE (β = 0.56, *p* < 0.001, 95% CI = [0.47, 0.63]), and CSE negatively predicted AB (β = −0.56, *p* < 0.001, 95% CI = [−0.55, −0.43]). The indirect effect of the path with CSE as the mediating variable was −0.27 (95% CI = [−0.33, −0.21]), accounting for 52.64% of the total effect; AM mediated between SC and AB, with SC positively predicting AM (β = 0.57, *p* < 0.001, 95% CI = [0.71, 0.94]) and AM negatively predicting AB (β = −0.11, *p* < 0.01, 95% CI = [−0.11, −0.03]). The indirect effect of the pathway with AM as a mediating variable was −0.05 (95% CI = [−0.09, −0.01]), which accounted for 10.91% of the total effect; ASH mediated between SC and AB, where SC negatively predicted ASH (β = −0.15, *p* < 0.01, 95% CI = [−0.22, −0.03]), and ASH positively predicted AB (β = 0.13, *p* < 0.001, 95% CI = [0.06, 0.20]). The indirect effect of the pathway with ASHs as a mediating variable was −0.02 (95% CI = [−0.04, −0.01]), accounting for 3.44% of the total effect.

#### 3.2.3. Chain Mediation of CSE, AM, and ASH

CSE and ASH acted as chain mediators between SC and AB, where CSE negatively predicted ASH (β = −0.16, *p* < 0.001, 95% CI = [−0.22, −0.04]). The indirect effect of the pathway with CSE and ASH as the mediating variables was −0.01 (95% CI = [−0.02, −0.01]), accounting for a total effect of 2.05%. AM and ASH acted as chain mediators between SC and AB, where AM negatively predicted ASH (β = −0.22, *p* < 0.001, 95% CI = [−0.17, −0.05]), with an indirect pathway effect of −0.01 (95% CI = [−0.03, −0.01]), with CSE and ASH as mediating variables, accounting for 2.74% of the total effect.

## 4. Discussion

On the basis of the COR perspective, this study examines the influence of school connection on junior high school students’ academic burnout and its mechanism. The results show that the greater the school connection level of junior high school students is, the lower their academic burnout. Hypothesis 1 is supported. Autonomous motivation, core self-evaluation, and academic self-obstruction not only play independent mediating roles in the relationship between school connection and academic burnout. Hypotheses 2~4 are supported. School connection can also influence academic burnout through the chain mediating effects of autonomous motivation, core self-evaluation, and academic self-obstruction, respectively. Hypotheses 5 and 6 are supported. This study has implications for the literature and for the theories on the causes of AB in middle school students.

### 4.1. School Connectedness and Academic Burnout

This study revealed that SC significantly predicted AB and that the higher middle school students’ SC levels were, the less affected they were by AB. The results also support the idea of COR that students engage in emotional consumption in school due to academic pressure, etc., and that SC, as a kind of resource, supplements wasted resources [12], thus, reducing the occurrence of AB. In addition, students with high SC are willing to learn more and actively participate in school activities, which promotes their academic, physical and mental development and effectively reduces the emergence of problematic behaviors [56]. When middle school students form good SCs, they may receive more support and help from teachers and peers, which not only meets the basic psychological needs of middle school students but also increases their positive psychology in coping with academic stress [57], thus, preventing the occurrence of negative consequences such as AB.

### 4.2. Mediating Role of Core Self-Evaluations

This study shows that CSE partially mediates the relationship between SC and AB. This mediating effect not only supports COR but is also consistent with evidence from previous research on the protective effect of high levels of CSE on AB [36]. When adolescents perceive low school belongingness, their CSE is likely to decrease, and positive SC positively predicts their CSE [58]. High levels of teacher support, peer support, and other positive relational connections help enhance students’ academic, interpersonal, and other self-efficacy [22], which leads them to affirm their own abilities and generate greater CSE. Second, students’ CSE helps reduce AB problems among middle school students. Students with greater CSE are more confident in facing the challenges of learning, resulting in more positive experiences and less AB [59]. Therefore, students can reduce their level of AB by increasing their CSE in a high-SC environment.

### 4.3. Mediating Role of Autonomous Motivation

AM partially mediates the relationship between SC and AB, a result that again supports PMF [19]. The support of teachers and peers in school can fulfill students’ three basic psychological needs—autonomy, competence, and belonging—and when students feel supported by their teachers, they are motivated to learn more autonomously [40]. When teachers provide an autonomous learning environment in which students’ various learning and exploratory activities are not constrained, students feel that their learning behaviors are self-determined [60]; i.e., the more support students feel and the greater their SC, the more learning autonomy motivation is promoted. Thus, good SC may positively impact students’ learning autonomy motivation. Therefore, the greater the SC of middle school students is, the more students can satisfy their basic psychological needs, experience more autonomy, and generate AM. In addition, when students’ AMs for learning are stronger, when students are interested in learning itself or are actively involved in learning activities, they have higher levels of engagement in learning and lower levels of AB [9]. COR also holds that AM is the key factor for students to maintain learning status, which can promote students to engage in positive behaviors to increase their level of psychological resources and realize self-resource replenishment, and individuals with high AM for learning show more positive learning behaviors during the learning process, which results in less AB [32].

### 4.4. Mediating Role of Academic Self-Handicapping

ASH mediates the relationship between SC and AB, i.e., SC reduces students’ ASH and enables them to view academic frustration in a positive light, which reduces the emergence of AB behaviors. This finding further supports psychological mediation theory and COR and is consistent with previous studies [19,44]. On the one hand, the more SC students feel in the school environment, the more positively they can face academic problems and inhibit or alleviate the occurrence of ASH problems [27]. On the other hand, students who suffer from various kinds of neglect and frustration in their school environments have a low level of SC and fail to meet their basic psychological needs exhibit more internalizing and externalizing behaviors, resulting in the development of ASH behaviors. [61] According to COR, which suggests that individuals’ own factors and environmental factors interact to influence their psychology and behaviors, students with high ASH levels may form poor study habits and develop AB [12]. Therefore, SC can reduce the level of students’ ASH and help adolescents achieve positive problem solving in their academic development, which in turn reduces the occurrence of AB.

### 4.5. Chain Mediation of Core Self-Evaluation, Autonomous Motivation, and Academic Self-Handicapping

SC inhibits the AB of middle school students through the chain mediating effect of CSE, AM, and ASH. This result supports the cumulative effect model of risk-protective factors, which can have a positive effect by weakening risk factors. When individuals’ self-protection systems are disorganized, their CSE is low or their AM is sufficient, they easily generate more positive emotions, and these positive factors reduce the role of risk factors and activate the individual protection system [27], reducing the level of ASH. Students who have lower evaluations of themselves, who do not believe that they can complete academic tasks, and who doubt their own abilities do not respond positively to difficulties in learning but rather respond in an avoidant manner, resulting in an increase in ASH behaviors [47]. Therefore, SC can increase students’ CSE, allowing them to believe in their learning abilities and have confidence in challenging tasks. An increase in CSE can reduce students’ ASH, which in turn reduces the problem of AB. AM encourages students to adopt appropriate learning strategies to cope with learning tasks [27], increases their likelihood of success, stimulates their enthusiasm for learning, and reduces the implementation of self-hindering strategies. Since higher AM for learning helps students maintain serious learning attitudes and persevere in completing learning tasks over a long period and has a maintaining and regulating effect on learning behaviors [39,62], individuals are less likely to develop ASH problems under AM. Therefore, middle school students’ SC during the learning process has a significant effect on whether they can learn autonomously, and students’ AM can further help them choose appropriate learning strategies to avoid ASH, which, ultimately, reduces the level of AB.

## 5. Limitations and Future Research

This study has the following shortcomings. First, this study collected data through a self-report questionnaire method, which has limitations such as high subjectivity and a lack of causal explanatory strength, and there are biases and inaccuracies in self-report data; for example, self-reports may be affected by social desirability or recall bias. In future studies, a variety of research methods should be used to examine the subjects in an integrated manner, such as multiple sources of information (e.g., teachers and parents), interview methods, and observation methods. Second, although previous theoretical and empirical studies provide a solid foundation for this study, the cross-sectional design still fails to reveal causal relationships among variables. Future research should consider the use of longitudinal studies to more effectively explain the essence of the relationship between SC and AB. Third, only factors related to the students themselves and factors related to the school were selected in this study to explore the relationship between SC and ABs. Family factors and the natural environment also have important impacts on autonomous learning [13]. Studying the regulatory role that parental educational involvement and natural support may play in the relationship mechanism between SC and AB is possible. For example, it is necessary to study whether students with high parental educational involvement have a weak negative correlation between SC and AB to determine whether parental educational involvement can play a moderating role. Basic psychological needs, coping styles, and social support have been shown to affect AB [30,63], and these factors should be added as potential mediating variables in the future to explore the effect of SC on AB to further enrich the understanding of this relationship. Fourth, the sample of this study included Chinese middle school students, and the representative sample was not strong; thus, the research results lack generalizability. Future studies may expand the scope of the subjects and survey students in more countries to explore the universality of the results. Finally, in future studies, researchers should seek to further reduce AB among middle school students by developing intervention programs for SC.

## 6. Conclusions and Implications for Practice

This study examines the causes of AB from the perspective of SC and reveals that CSE, AM, and ASH play multiple mediating roles, which provides some preliminary suggestions for schools to implement measures to alleviate the problems and risks of students’ AB.

First, SC has a protective effect on junior high school students’ AB. A school should create a positive and democratic school atmosphere and provide more interpersonal support among teachers and students. Only in such a supportive and safe school atmosphere can students feel treated fairly, that their inner emotions and needs are satisfied, that they feel less pressure, and that their sense of belonging to the school is enhanced, thus, reducing the occurrence of AB. Teachers should provide more support to students, meet the relationship needs of students, enhance the close connection between students and schools, and meet the needs of individual belonging and love. In addition, peer mentoring programs or workshops focused on social-emotional learning can be developed to increase peer support. Teachers can also identify students’ ABs in a timely manner through daily observations or questionnaires and effectively intervene to create a supportive environment that promotes both academic success and emotional health.

Second, SC can affect students’ CSE, thus, reducing the risk of AB. On the one hand, schools can organize a variety of classroom activities so that students can enhance their understanding, reflect on and improve in a discussion atmosphere, promoting the improvement of students’ CSE. On the other hand, teachers should respect students’ emotional experiences; build a democratic, harmonious and equal relationship between teachers and students; listen to students’ comments; allow students to question and interpret; carry out vivid self-evaluations and mutual evaluations; and create a harmonious and relaxed evaluation atmosphere so that students can constantly educate themselves in the learning process, build self-confidence, and improve their CSE ability. Therefore, by creating a vivid situation, students can have a successful experience, and teachers should explore and gradually cultivate, respect, and encourage students’ CSE, magnify students’ self-evaluation abilities, make them fully feel the fun of self-evaluation, and encourage students to dare to CSE. Teachers should find students’ learning gains and shortcomings, allow them to experience the joy of success in a timely manner, and gradually develop good habits of CSE to reduce AB.

Third, SC can reduce AB problems by improving students’ AM. First, schools should organize learning activities to stimulate students’ learning interest and cultivate good learning habits, maintain their motivation and passion for learning, and boldly try new learning areas and methods. They should also teach effective learning methods and strategies to help students discover the rules and nature of learning and master self-management skills so that learners can better improve their ability to independently learn. Second, in teaching, teachers should foster a self-sufficient atmosphere for students, motivate them to think about and solve issues on their own, and support the development of their motivation. Teachers should change the traditional teacher-oriented education mode, which ignores students’ learning subjectivity, and adopt an independent support style to stimulate the AM of left-behind children and reduce AB problems. Moreover, teachers should pay more positive attention and support to students facing ABs so that they can reduce their negative academic emotions by improving their internal motivation. By stimulating students’ AM, they can increase their enthusiasm for learning, increase their autonomy in learning, and reduce the probability of AB.

Fourth, SC can also affect AB by reducing the occurrence of students’ ASH behaviors. In terms of schools, teachers should encourage students to be more self-affirming and encourage their inner self-growth instead of emphasizing the role of others’ evaluations in their self-cognition, improving their self-efficacy, and helping them improve their learning strategies to avoid the occurrence of ASH. On the family side, to avoid ASH caused by unstable self-esteem, parents should improve the way they praise their children, provide unconditional and consistent care to the child and provide a basic sense of self-esteem; at the same time, parents should give praise that is consistent with the child’s performance and be clear about the reasons behind the praise. These strategies can be used to reduce the occurrence of ASH, help students actively face the difficulties they encounter in learning, and reduce AB.

## Figures and Tables

**Figure 1 behavsci-14-01077-f001:**
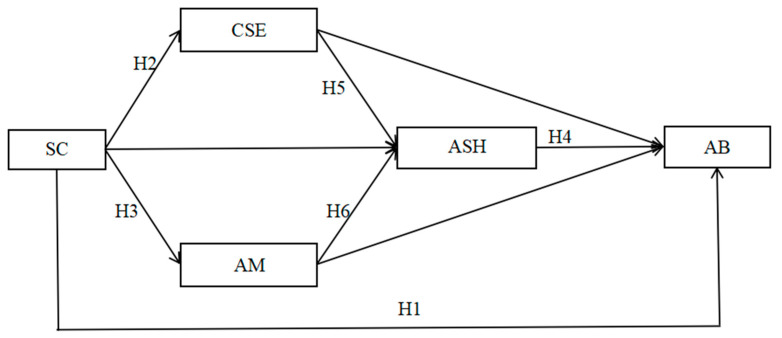
Multiple sequential mediation effects model of school connectedness and academic burnout. Note: SC = school connectedness; AB = academic burnout; CSE = core self-evaluation; AM = autonomous motivation; ASH = academic self-handicapping; same below.

**Figure 2 behavsci-14-01077-f002:**
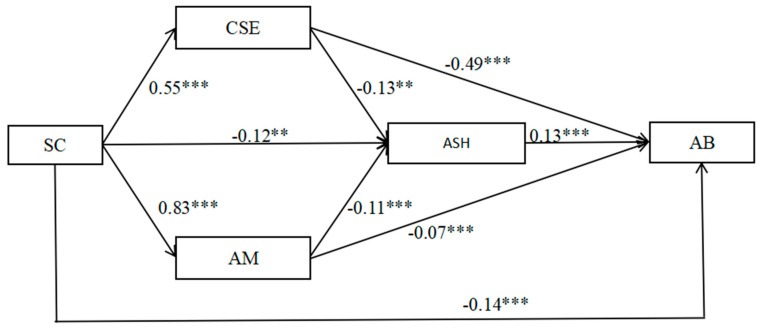
Multiple sequential mediation models of school connectedness, core self-evaluations, autonomous motivation, academic self-handicapping, and academic burnout. ** *p* < 0.01, *** *p* < 0.001.

**Table 1 behavsci-14-01077-t001:** Reliability Statistics for the Scale.

Scale/Factors	Cronbach’s α	SE α	BootLLCI, BootULCI	ω
1. PAAM	0.89	0.01	[0.88, 0.91]	0.90
2. PAAM (feelings of rejection)	0.73	0.03	[0.66, 0.78]	0.74
3. PAAM (feelings of belonging to the school)	0.89	0.01	[0.87, 0.90]	0.89
4. ABS	0.87	0.01	[0.85, 0.89]	0.88
5. ABS (emotional exhaustion)	0.78	0.01	[0.74, 0.81]	0.78
6. ABS (academic detachment)	0.71	0.02	[0.65, 0.75]	0.71
7. ABS (low achievement)	0.77	0.02	[0.72, 0.81]	0.77
8. CSE	0.87	0.01	[0.85, 0.89]	0.88
9. ASRQ	0.91	0.01	[0.89, 0.93]	0.91
10. ASHS	0.91	0.01	[0.89, 0.93]	0.92
11. ASHS (academic self-handicapping)	0.87	0.01	[0.83, 0.89]	0.87
12. ASHS (claimed academic self-handicapping)	0.84	0.01	[0.79, 0.86]	0.84

**Table 2 behavsci-14-01077-t002:** Descriptive statistics and correlation analysis results for each variable.

Variant	M	SD	1	2	3	4	5
1. SC	4.57	0.86	1				
2. AB	2.62	0.73	−0.63 **	1			
3. CSE	3.53	0.84	0.58 **	−0.78 **	1		
4. ASH	1.68	0.67	−0.37 **	0.43 **	−0.34 **	1	
5. AM	5.56	1.25	0.58 **	−0.55 **	0.47 **	−0.38 **	1
6. Gender ^0^			0.05	−0.09	0.12 *	0.01 *	0.01
7. Grade ^1^			−0.11 *	0.10 *	−0.12 *	−0.02	−0.10 *
8. Grade ^2^			−0.05	0.12 *	−0.07	0.06	−0.03

Note: * *p* < 0.05, ** *p* < 0.01; gender is coded via dummy coding: males are controls, Gender ^0^ = girls; grade level is coded via dummy coding: 7th grade is control, Grade ^1^ = 8th grade, Grade ^2^ = 9th grade; same below.

**Table 3 behavsci-14-01077-t003:** Results of regression analysis.

Variant	CSE			AM			ASH			AB		
	*β*	*SE*	*t*	*β*	*SE*	*t*	*β*	*SE*	*t*	*β*	*SE*	*t*
Gender ^0^	0.14	0.06	2.16	−0.07	0.10	−0.75	0.17	0.06	2.94	−0.03	0.04	−0.70
Grade ^1^	−0.15	0.08	−1.83	−0.13	0.12	−1.00	−0.13	0.07	−1.80	−0.05	0.05	1.08
Grade ^2^	−0.13	0.08	−1.66	−0.05	0.12	−0.845	0.00	0.07	0.09	0.11	0.05	2.37
SC	0.55	0.04	13.85 ***	0.83	0.06	13.83 ***	−0.12	0.04	−2.66 **	−0.14	0.03	−4.29 ***
CSE							−0.13	0.04	−3.00 **	−0.49	0.03	−15.34 ***
AM							−0.11	0.03	−3.84 ***	−0.07	0.02	−3.57 ***
ASH										0.13	0.03	3.93 ***
*R* ^2^	0.36			0.34			0.22			0.68		
*F*	54.83 ***			50.33 ***			18.27 ***			121.90 ***		

Note: ** *p* < 0.01, *** *p* < 0.001.

**Table 4 behavsci-14-01077-t004:** Analysis of indirect effects and comparison of effects.

Effect (Scientific Phenomenon)	Efficiency Value	[BootLLCI, BootULCI]	Percentage of Total Effect (ab/c)
Total indirect effect	−0.38	[−0.45, −0.31]	72.61%
Indirect effect 1	−0.27	[−0.33, −0.21]	52.35%
Indirect effect 2	−0.06	[−0.10, −0.02]	12.25%
Indirect effect 3	−0.01	[−0.03, −0.01]	3.37%
Indirect effect 4	−0.01	[−0.02, −0.01]	2.03%
Indirect effect 5	−0.01	[−0.02, −0.01]	2.59%

Note: Indirect effect 1 is school connectedness → core self-evaluation → academic burnout; indirect effect 2 is school connectedness → autonomous motivation → academic burnout; indirect effect 3 is school connectedness → academic self-handicapping → academic burnout; indirect effect 4 is school connectedness → core self-evaluation → academic self-handicapping → academic burnout; and indirect effect 5 is school connectedness → autonomous motivation → academic self-handicapping → academic burnout depression. The total indirect effect was the sum of the above five indirect effects. c is the total effect of X on Y, ab is the intermediary effect through the intermediary variable M, and the ratio of the intermediary effect to the total effect is ab/c.

## Data Availability

The raw data supporting the conclusions of this article will be made available by the authors upon request.

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
