# Peer review of "School Connectedness and Academic Burnout in Middle School Students: A Multiple Serial Mediation Model"

_behavsci, 2024, doi:10.3390/bs14111077_

Round 1
Reviewer 1 Report
Comments and Suggestions for Authors
A concise introduction enables the reader to understand the research problem.
I would like to know more about the academic system in China to explain academic burnout in depth.
The instrument's development and content (ASQR test) must be sufficiently described and referenced. I would have liked to read an explicit rationale about your methodological choices.
Overall, the research process and the results could be explained more thoroughly to make the study more transparent and informative.
Findings
The reporting of results, the tables, and the associated data should be considered.
The conclusion does not provide sufficient information to allow a reader to understand this research's contribution.
The author (s) need to mention ethical issues for their study.
In general, the English in the present manuscript is of publication quality and requires minor improvement.
Comments on the Quality of English LanguageIn general, the English in the present manuscript is of publication quality and requires minor improvement.
Author Response
Comments 1: A concise introduction enables the reader to understand the research problem |
Response 1:Thank you for your comments. We have reduced the introduction content in the article, such as “AB can cause anxiety, sadness, and low self-esteem (Fiorilli et al., 2017; Dong et al., 2021). On the one hand, there are externalizing problems such as academic procrastination (Madigan & Curran, 2021), dropping out of school, and declining academic performance (Bask & Salmela-Aro, 2013). On the other hand, in terms of behavior, AB is detrimental to those who experience it and has emerged as a typical psychological issue among middle school students./SC positively affects individuals’ learning and physical and mental health development and is a direct protective factor for students' subjective well-being (Wu et al., 2022) and academic achievement (Yin & Bai, 2022)” |
Comments 2: I would like to know more about the academic system in China to explain academic burnout in depth. |
Response 2: Thank you for your valuable question. At present, China is implementing the secondary school entrance examination diversion policy, the main purpose of which is to divide students into two groups of talents with different development directions. Students with ideal performance can enter ordinary high schools and key high schools to study, and such students will become highly educated talents focused on training by the education department, while those with poor performance can only be diverted to vocational schools and technical secondary schools. This kind of students will become the national key training of high-tech talents. Most people regard the high school entrance examination as the first watershed of life, so the students in this period have greater learning pressure, will face many changes in learning, will produce a variety of bad academic emotions, and eventually lead to a variety of learning problems, and academic burnout is one of the most common. Academic burnout is that students have great academic pressure in the learning process and have no interest in learning, leading to boredom and burnout of learning. |
Comments 3: The instrument's development and content (ASQR test) must be sufficiently described and referenced. l would have liked to read an explicit rationale about your methodological choices |
Response 3: Thank you for your comments. ASQR is short for Academic Self-Regulation Questionnaire. We use the self-regulation dimension of the Academic Self-regulation Questionnaire to measure students' autonomous motivation. This questionnaire is a revision of the original English Learning self-regulation Questionnaire (Black & Deci, 2000) by scholar Chen Xuelian on the basis of literature. The resulting self-regulation questionnaire is divided into two dimensions: self-regulation and external control. There are 8 questions each. Referring to previous Chinese studies, most of them use the autonomous regulation dimension of this scale to measure autonomous motivation, which shows good reliability and validity in the study of measuring autonomous motivation, and is more suitable for measuring the autonomous motivation of Chinese junior middle school students. For example, Jiang Tiantian et al. (2023) wrote in describing the measurement tool of autonomous motivation that "the autonomous regulation dimension in the academic self-regulation questionnaire compiled by Black and Deci (2000) and revised by Chen Xuelian (2007) was used to measure students' autonomous motivation". Therefore, we chose this scale to measure autonomous motivation. We have made changes to the description of the scale in the paper, please refer to the "Measures" (pp.7; line 313-323) section, where we have highlighted these modifications in red [This questionnaire is a revision of the original English Learning self-regulation Questionnaire (Black & Deci, 2000) by scholar Chen Xuelian on the basis of literature. Finally, the self-regulation questionnaire was divided into the self-regulation dimension and the external control dimension with 8 questions, respectively. Referring to previous Chinese studies, most of them use the autonomous regulation dimension of this scale to measure autonomous motivation (for example, "I think actively participating in class is a good way to improve my learning"), which shows good reliability and validity in the research on measuring autonomous motivation(Jiang et al.,2023; Lang et al., 2022), and is more suitable for measuring the autonomous motivation of Chinese junior high school students.] |
Comments 4: Overall., the research process and the results could be explained more thoroughly to make the study more transparent and informative. |
Response 4: Thank you for your comments. According to your suggestions, we added interpretation and analysis methods to the research process, and separately described the relationship between variables in the results part to make the process and results more clear. Please refer to the "Data Analyses" (pp.7; line 337-348) and "Results" (pp.9; line 372-375) section, where we have highlighted these modifications in red [In Models 1 and 2, we regressed the control variables SC on CSE and AM, respectively; in Model 3, we regressed the control variables SC, CSE, and AM on ASH; and in Model 4, we regressed the control variables SC, CSE, AM, and ASH on AM. The results of the whole regression equation were significant.../Controlling for class, gender, and grade, SC was significantly correlated with AB (see Table 3, Figure 2). Specifically, SC was significantly and negatively correlated with AB (β =-0.17, p<0.001), with a 95% confidence interval of [-0.20, -0.07] and an adjusted R2 =0.69, F=79.61, p<0.001 for the model.] |
Comments 5: The reporting of results, the tables, and the associated data should be considered. The conclusion does not provide sufficient information to allow a reader to understand this research's contribution. |
Response 5: Thank you for your comments. We have enriched and explained the results, tables and related data to make the results more clearly; The description of the conclusions is added to make readers more clearly understand the conclusions and contributions of this research. Please refer to the "Results" (pp.9; line 371-410) and "Discussion" (pp.10; line 415-423) section, where we have highlighted these modifications in red [After controlling for the main effects of class, gender, and grade on students, the mediating variables—CSE, AM, and ASH—were added to the model to obtain the path model shown in Figure 2.../Based on the COR perspective, this study examines the influence of school connection on junior high school students' academic burnout and its mechanism. The results show that the higher the school connection level of junior high school students, the lower their academic burnout, hypothesis 1 is supported.] |
Comments 6: The author (s)need to mention ethical issues for their study |
Response 6: Thank you for your comments. We have added theoretical ethical issues to the text. Please refer to the "Materials and Methods" (pp.5-6; line 259-261) section, where we have highlighted these modifications in red [This study was conducted in accordance with the Declaration of Helsinki, and approved by the Ethics Committee of the Faculty of Education, Henan Normal University] |
Comments 7: In general, the English in the present manuscript is of publication quality and requires minor improvement. |
Response 7: Thank you for your comments. We have touched up some English in the article to make it read more smoothly |
Reviewer 2 Report
Comments and Suggestions for Authors
Abstract: Τhe phrase “underlying mechanisms of burnout” is a bit unclear. Authors might consider rephrasing it to “mediating mechanisms of burnout”. I would also suggest replacing the sentence "revealed that" with "found that", because it has already been used in the second line. The authors could avoid redundant in study results, providing a concrete example of how these results could influence educational practices to enhance the impact of their conclusion as well.
Introduction: The introduction effectively sets the stage for the study by discussing the problem of academic burnout among Chinese middle school students. However, the authors introduce self- evaluations (CSE), autonomous motivation (AM) and academic self-handicapping (ASH) without explaining why each of these variables is important for understanding the relationship between school connectedness (SC) and academic burnout (AB).
Theoretical framework: I like the way the hypotheses emerge from the theoretical framework.
The Present Study: To be more meaningful to the readers, it might help to briefly explain why gender and grade level are significant as control variables.
Materials and Methods: The term “no nesting” may need clarification. I would also like to read further details about why “two rounds were conducted 2 weeks apart”. The term “effective rate” (87.0%) could be better explained (how it has been calculated, why this rate is considered sufficient for the study's objectives) as well.
Data Analyses: It would be helpful to specify here which model from the PROCESS macro you used (instead of 3.2. Structural Model).
Results: The tables and figures are well-detailed and accurate, effectively supporting the findings presented in the study.
Discussion: I find the author’s division of the discussion into separate sections effective.
Limitations and Future Research: It might be beneficial to elaborate briefly on how self-report questionnaires might affect the findings, such as potential biases or inaccuracies in self-reported data (e.g., social desirability or recall bias).
It may be helpful to provide examples of how the inclusion of additional factors (i.e., family factors and the natural environment) could be operationalized in future research.
Conclusion and Implications for Practice: The conclusion effectively highlights the protective role of school connectedness (SC) in mitigating academic burnout (AB) among middle school students.
The authors could also include specific examples of initiatives or programs that schools could implement, recommending peer mentoring programs or workshops that focus on social-emotional learning. In addition, they could propose that teachers can be equipped with the tools and resources to recognize signs of academic burnout and intervene effectively, creating a supportive environment that nurtures both academic success and emotional health.
Author Response
Comments 1: Abstract: The phrase “underlying mechanisms of burnout"is a bit unclear. Authors might consider rephrasing it to "mediating mechanisms of burnout". l would also suggest replacing the sentence "revealed that" with "found that", because it has already been used in the second line.The authors could avoid redundant in study results, providing a concrete example of how these results could influence educational practices to enhance the impact of their conclusion as well. |
Response 1:Thank you for your comments. In the summary, we have replaced "underlying mechanisms of burnout" with "mediating mechanisms of burnout"; Replace "revealed that" with "found that"; As well as the conclusion of the study, the examples of educational practice are added, which are enriched and improved. Please refer to the "Abstract" (pp.1; line 10-11,16-20)section, where we have highlighted these modifications in red [mediating mechanisms of burnout/found/Therefore, educators can pay attention to students' emotional needs from the school factors of junior high school students, increase the level of school connectedness, and consciously help students cultivate positive psychological factors such as autonomous motivation and core self-evaluation, reduce academic self-handicapping, enhance students' learning pleasure, and alleviate junior high school students' academic burnout.] |
Comments 2: Introduction: The introduction effectively sets the stage for the study by discussing the problem of academic burnout among Chinese middle school students. However, the authors introduce self-evaluations (CSE), autonomous motivation (AM) and academic self-handicapping (ASH)without explaining why each of these variables is important for understanding the relationship between school connectedness (SC) and academic burnout (AB). |
Response 2: Thank you for your comments.. Conservation of resources theory (COR) holds that individuals tend to adopt good coping strategies when their resources are sufficient, which provides sufficient guarantees for individuals to cope with AB. If individual resources are limited and difficult to cope with, persistent resource threats and depletion will lead to AB (Wang et al., 2023). SC provides a supportive environment that facilitates the replenishment of individuals’ personal capacities, which helps them maintain resources (Halbesleben et al., 2014). CSE enables individuals to believe in their own abilities and positively face learning and life, thus enriching the individual’s store of psychological resources needed for the learning process (Zhou et al., 2014). AM maintains students' learning state to achieve self-resource retention (Reeve, 2013). All three of these factors can effectively replenish the resources lost by individuals. In contrast, ASH, as a negative coping style, inhibits the achievement of challenging goals and personal growth enhancement (Zhang et al., 2021), preventing individuals from building general resources. We have described the possible role of these variables in the relationship between SC and AB and added them to the introduction. Please refer to the " Introduction" (pp.2; line 75-83) section, where we have highlighted these modifications in red [SC provides a supportive environment that facilitates the replenishment of individuals’ personal capacities, which helps them maintain resources (Halbesleben et al., 2014)...] |
Comments 3: Theoretical framework: l like the way the hypotheses emerge from the theoretical framework. |
Response 3: Thank you for your support in writing the theoretical framework. |
Comments 4: The Present Study: To be more meaningful to the readers, it might help to briefly explain whygender and grade level are significant as control variables. |
Response 4: Thank you for your comments. As you said, gender and grade are related to the level of academic burnout, and the level of male academic burnout is generally higher than that of female, and the level of academic burnout will increase with the growth of grade. To reduce the influence of both on academic burnout and make the result more convincing, both are controlled as control variables. We have made explanations and modifications in the paper. Please refer to the "The Present Study" (pp.5; line236-240) section, where we have highlighted these modifications in red [Previous studies have shown that gender and grade are related to the level of AB, and the level of male AB is generally higher than that of female, and the level of AB will increase with the growth of grade (He et al., 2023). To reduce the impact of both on AB and make the results more authentic, both of them are controlled as control variables.] |
Comments 5: Materials and Methods: The term “no nesting" may need clarification. l would also like to read further details about why “two rounds were conducted 2 weeks apart". The term "effective rate 87.0%)could be better explained (how it has been calculated, why this rate is considered sufficient for the study's objectives) as well. |
Response 5: Thank you for your comments and questions. â‘ The paper states that "no nesting used in this study", in order to show that there is no multilevel hierarchical structure of the data studied. This term is not used properly and we have amended it in this article. Please refer to the "Materials and methods" (pp.5; line 244) section where these modifications are highlighted in red [Nested methods were not used in this study] â‘¡ "Two weeks apart" is divided into two time points to collect questionnaires in order to reduce the impact of common method bias, which can reduce the fatigue of students in filling out questionnaires. The research shows that common method deviation control can be divided into two kinds: program control and system control. Using a program-controlled approach, mainly for the study process, one possible measure is to measure the variables separately, for example by introducing a time lag between the measurements of the predictor and the standard variables to create a time separation and reduce the measurement error (Podsakoff et al.,2003). With reference to previous studies, it is believed that common method bias can be programmed by sending questionnaires at two time points (Tan et al.,2022; Deng et al., 2018). For example, in Tan Chunping's study, "Two online data collections will be conducted one month apart in April and June 2021 for procedural control of common methodology bias." [1]Tan, C., Tian, R., & Zhang, Y.(2022). Research on the influence mechanism and gender differences of family-supportive supervisor behavior on employee workplace well - being. Journal of Management, (3), 144-158. https://doi.org/10.19808/j.cnki.41-1408/f.2022.0031 [2]Podsakoff, P. M., MacKenzie, S. B., Lee, J. Y., & Podsakoff, N. P. (2003). Common method biases in behavioral research: A critical review of the literature and recommended remedies. Journal of Applied Psychology, 88(5), 879. https://doi.org/10.1037/0021-9010.88.5.879 â‘¢ Efficiency refers to the efficiency of the questionnaire, generally speaking, it is obtained by calculating the proportion of the number of people who effectively fill out the questionnaire to the total number of people participating in the survey. The specific calculation formula is: efficiency = number of valid entries/total number of participants *100%. If the efficiency rate is too low, such as less than 60%, it may mean that there is a problem with the questionnaire design, or there is a problem in the survey implementation process, resulting in a lot of questionnaire data cannot be used. In contrast, if the efficiency rate is higher, such as 80% or 90%, it usually indicates that the questionnaire design and survey implementation are successful. We have made some changes in the article Please refer to the "Materials and Methods" (pp.6; line 251-256) section, where we have highlighted these modifications in red [After eliminating invalid questionnaires such as incomplete answers and irregular answers(incomplete answers meant that most of the questions were not answered, such as the questionnaire missing at least one variable, while the questionnaire missing only a few questions was retained, and the missing value was processed by the maximum likelihood method), 394 valid questionnaires were obtained , and the questionnaire efficiency was 87.5%. ] |
Comments 6: Data Analyses: It would be helpful to specify here which model from the PROCESS macro you used (instead of 3.2. Structural Model). |
Response 6: Thank you for your comments. Based on your suggestion, we have added the data analysis section to the description of which model is used. Please refer to the "Data Analyses" (pp.7; line 337-339) section, where we have highlighted these modifications in red [Based on the correlation analysis, model 80 of PROCESS, an SPSS plug-in provided by Hayes (2012), was used to construct multiple serial mediation models, and regression analysis was performed for models 1 to 4.] |
Comments 7: Results: The tables and figures are well-detailed and accurate, effectively supporting the findings presented in the study. |
Response 7: Thank you for your approval of the way the results are presented. |
Comments 8: Discussion:I find the author's division of the discussion into separate sections effective |
Response 8: Thank you for your approval of writing the discussion section. |
Comments 9: Limitations and Future Research: lt might be beneficial to elaborate briefly on how self-report questionnaires might affect the findings, such as potential biases or inaccuracies in self-reported data(e.q., social desirability or recall bias). It may be helpful to provide examples of how the inclusion of additional factors (i.e.. family factors and the natural environment) could be operationalized in future research. |
Response 9: Thank you for your comments. We describe and refine how self-reporting affects the results in limitations and future studies, and suggest ways to add family factors and the natural environment to the study in future studies. Please refer to the "Limitations and Future Research" (pp.12-13; line 515-518, 527-533) section, where we have highlighted these modifications in red [First, this study collected data through a self-report questionnaire method, which has limitations such as high subjectivity and a lack of causal explanatory strength,and there are biases and inaccuracies in self-report data, for example, self-report may be affected by social desirability or recall bias./Family factors and natural environment also have an important impact on autonomous learning (Wang et al., 2023). It is possible to study the regulatory role that parental educational involvement and natural support may play in the relationship mechanism between SC and AB. For example, it is necessary to study whether students with high parental educational involvement have a weak negative correlation between SC and AB. To determine whether parental educational involvement can play a moderating role.] |
Comments 10: Conclusion and implications for Practice: The conclusion effectively highlights the protective role of school connectedness (SC) in mitigating academic burnout (AB) among middle school students. |
Response 10: Thank you for your recognition of the conclusions and practical implications of this study |
Comments 11: The authors could also include specific examples of initiatives or programs that schools could implement, recommending peer mentoring programs or workshops that focus on social-emotional learning. In addition, they could propose that teachers can be equipped with the tools and resources to recognize signs of academic burnout and intervene effectively, creating a supportive environment that nurtures both academic success and emotional health. |
Response 11: Thank you for your comments. We very much agree with your point of view, and in the conclusion and practical inspiration to explain and improve. Please refer to the "Conclusion and Implications for Practice" (pp.13; line 556-560) section, where we have highlighted these modifications in red [In addition, peer mentoring programs or workshops focused on social-emotional learning can be developed to increase peer support. Teachers can also identify students' AB in a timely manner through daily observation or questionnaire filling, and effectively intervene to create a supportive environment that promotes both academic success and emotional health.] |
Reviewer 3 Report
Comments and Suggestions for Authors
In my opinion, the study is very interesting and worth publishing. Indeed, the article is well structured, the method used is congruent with the purpose of the study and the results are clearly presented.
1. Summary of the Article
The article addresses the relationship between school connectedness and academic burnout by taking Chinese middle school stutudents as research subjects. The authors construct a multi-sequential mediation model in accordance with COR and PMF theories, and explore in depth the impact of SC on AB, as well as the multiple sequential mediating roles of CSE, AM and ASH in the aforementioned relationships. Empirical data are presented to support their findings.
2. Overall Assessment
In my opinion, the study is very interesting and worth publishing. Indeed, the article is well structured, the method used is congruent with the purpose of the study and the results are clearly presented.
General comments:
- The topic is relevant and of great interest in the field of education.
- The structure of the article is clear and logical.
- The research objectives are well defined.
3. Strengths of the Article
- Relevance of the Topic: The connection between school connectedness and academic burnout is a critical area that deserves attention.
- Methodology: The methodology used is adequate and described in sufficient detail.
- Results: The results are interesting and bring new information to the field.
4. Areas for Improvement
- Literature Review: The literature review could be expanded to include recent studies from other countries that address the question under investigation.
- Discussion: Although the arguments are adequate, the thread I think would be clearer if a paragraph was included at the beginning where it was made clear which of the 6 hypotheses that have been formulated have been confirmed/refuted.
- Limitations: The limitations do not take into account that the findings are not generalisable given that a sample of Chinese middle school students was used. It would be interesting to propose as a future line of research to replicate this research with students from other countries.
Author Response
Comments 1: Literature Review: The literature review could be expanded to include recent studies from other countries that address the question under investigation. |
Response 1:Thank you for your comments. According to your suggestion, we have added some recent and foreign studies to the literature review to demonstrate the relationship between variables. Please refer to the “Theoretical Background and Hypothesis Development” section (pp3-4; line122-123, 160-162, 180-183), where in the revised manuscript this change can be found [Among the factors that influence AB, SC has been shown to have an effect on it (Gao et al., 2024)./Students with positive CSE are more likely to use their psychological resources and act proactively in dealing with challenging situations, which can increase their burnout (PaloÈ™, 2024)./It has also been suggested that students who perceived their teachers as providing more autonomy support experienced more autonomous learning motivation which, in turn, led to more frequent flow experiences in learning and subsequently to less burnout(Ljubin-Golub et al., 2020). ] |
Comments 2: Discussion: Although the arguments are adequate, the thread l think would be clearer if a paragraph was included at the beginning where it was made clear which of the 6 hypotheses that have been formulated have been confirmed/refuted. |
Response 2: Thank you for your comments. We add a description at the beginning of the discussion, which shows which hypotheses of this study have been confirmed by this study. Please refer to the "Discussion" (pp.10-11; line 415-423) section, where we have highlighted these modifications in red [Based on the COR perspective, this study examines the influence of school connection on junior high school students' academic burnout and its mechanism. The results show that the higher the school connection level of junior high school students, the lower their academic burnout, hypothesis 1 is supported; Autonomous motivation, core self-evaluation and academic self-obstruction not only play an independent mediating role in the relationship between school connection and academic burnout, hypothesis 2~4 is supported; School connection can also influence academic burnout through the chain mediating effects of autonomous motivation, core self-evaluation and academic self-obstruction, respectively. Hypothesis 5,6 is supported.] |
Comments 3: Limitations. The limitations do not account for that the findings are not general is able given that a sample of Chinese middle school students was used. It would be interesting to propose as a future line of research to replicate this research with students from other countries |
Response 3: Thank you for your comments. In Limitations and Future Research, we added the limitations of the study sample and the direction of future efforts We have made changes to the description of the scale in the paper, please refer to the "Limitations and Future Research" (pp.12; line 515-518) section, where we have highlighted these modifications in red [Fourth, The sample of this study is Chinese middle school students, and the representative sample is not strong, so the research results lack of generalization. Future studies may expand the scope of the subjects and survey students in more countries to explore the universality of the results.] |